# REINFORCEMENT LEARNING WITH STRUCTURED HIERARCHICAL GRAMMAR REPRESENTATIONS OF ACTIONS

## ABSTRACT

From a young age we learn to use grammatical principles to hierarchically combine words into sentences. Action grammars is the parallel idea, that there is an underlying set of rules (a grammar) that govern how we hierarchically combine actions to form new, more complex actions. We introduce the Action Grammar Reinforcement Learning (AG-RL) framework which leverages the concept of action grammars to consistently improve the sample efficiency of Reinforcement Learning agents. AG-RL works by using a grammar inference algorithm to infer the action grammar of an agent midway through training. The agent's action space is then augmented with macro-actions identified by the grammar. We apply this framework to Double Deep Q-Learning (AG-DDQN) and a discrete action version of Soft Actor-Critic (AG-SAC) and find that it improves performance in 8 out of 8 tested Atari games (median +31%, max +668%) and 19 out of 20 tested Atari games (median +96%, maximum +3,756%) respectively without substantive hyperparameter tuning. We also show that AG-SAC beats the model-free state-of-the-art for sample efficiency in 17 out of the 20 tested Atari games (median +62%, maximum +13,140%), again without substantive hyperparameter tuning.

## 1 INTRODUCTION

Reinforcement Learning (RL) has made great progress in recent years, successfully being applied to settings such as board games (Silver et al., 2017), video games (Mnih et al., 2015) and robot tasks (OpenAI et al., 2018). Some of this advance is due to the use of deep learning techniques to avoid induction biases in the mapping of sensory information to states. However, widespread adoption of RL in real-world domains has remained limited primarily because of its poor sample efficiency, a dominant concern in RL (Wu et al., 2017), and complexity of the training process that need to be managed by various heuristics.

Hierarchical Reinforcement Learning (HRL) attempts to improve the sample efficiency of RL agents by making their policies to be hierarchical rather than single level. Using hierarchical policies can lead not only to faster learning, but also ease human understanding of the agent's behaviour – this is because higher-level action representations are easier to understand than low-level ones (Beyret et al., 2019). Identifying the right hierarchical policy structure is, however a non-trivial task (Osa et al., 2019) and so far progress in hierarchical RL has been slow and incomplete, as no truly scalable and successful hierarchical architectures exist (Vezhnevets et al., 2016). Not surprisingly most state-of-the-art RL agents at the moment are not hierarchical.

In contrast, humans, use hierarchical grammatical principles to communicate using spoken/written language (Ding et al., 2012) but also for forming meaningful abstractions when interacting with various entities. In the language case, for example, to construct a valid sentence we generally combine a noun phrase with a verb phrase (Yule, 2015). Action grammars are an analogous idea proposing there is an underlying set of rules for how we hierarchically combine actions over time to produce new actions. There is growing neuroscientific evidence that the brain uses similar processing strategies for both language and action, meaning that grammatical principles are used in both neural representations of language and action (Faisal et al., 2010; Hecht et al., 2015; Pastra & Aloimonos, 2012). We hypothesise that using action grammars would allow us to form a hierarchical action represen-

tation that we could use to accelerate learning. Additionally, in light of the neuroscientific findings, hierarchical structures of action may also explain the interpretability of hierarchical RL agents, as their representations are structurally more similar to how humans structure tasks. In the following we will explore the use of grammar inference techniques to form a hierarchical representation of actions that agents can operate on. Just like much of Deep RL has focused on forming unbiased sensory representations from data-driven agent experience, here we explore how data-driven agent experience of actions can contribute to forming efficient representations for learning and controlling tasks.

Action Grammar Reinforcement Learning (AG-RL) operates by using the observed actions of the agent within a time window to infer an action grammar. We use a grammar inference algorithm to substitute repeated patterns of primitive actions (i.e. words) into temporal abstractions (rules, analogous to a sentence). Similarly, we then replace repeatedly occurring rules of temporal abstractions with higher-level rules of temporal abstractions (analogous to paragraphs), and so forth. These extracted action grammars (the set of rules, rules of rules, rules of rules of rules, etc) is appended to the agent's action set in the form of macro-actions so that agents can choose (and evaluate) primitive actions as well as any of the action grammar rules. We show that AG-RL is able to consistently and significantly improve sample efficiency across a wide range of Atari settings.

## 2    RELATED WORK

Our concept of action grammars that act as efficient representations of actions is both related to and different from existing work in the domain of Hierarchical Control. Hierarchical control of temporally-extended actions allows RL agents to constrain the dimensionality of the temporal credit assignment problem. Instead of having to make an action choice at every tick of the environment, the top-level policy selects a lower-level policy that executes actions for potentially multiple time-steps. Once the lower-level policy finishes execution, it returns control back to top-level policy. Identification of suitable low level sub-policies poses a key challenge to HRL.

Current approaches can be grouped into three main pillars: Graph theoretic (Hengst, 2002; Mannor et al., 2004; Simsek et al., 2004) and visitation-based (Stolle & Precup, 2002; Simsek et al., 2004) approaches aim to identify "bottlenecks" within the state space. Bottlenecks are regions in the state space which characterize successful trajectories. Our work, on the other hand, identifies patterns solely in the action space and does not rely on reward-less exploration of the state space. Furthermore, our action grammar framework defines a set of macro-actions as opposed to full option-specific sub-policies. Thereby, it is less expressive but more sample-efficient to infer.

Gradient-based approaches, on the other hand, discover parametrized temporally-extended actions by iteratively optimizing an objective function such as the estimated expected value of the log likelihood of the observed data under the current policy with respect to the latent variables in a probabilistic setting (Daniel et al., 2016) or the expected cumulative reward in a policy gradient context (Bacon et al., 2017; Smith et al., 2018). Our action grammar induction, on the other hand, infers patterns without supervision solely based on a compression objective. The resulting parse tree provides an interpretable structure for the distilled skill set.

Furthermore, recent approaches in macro-action discovery (Vezhnevets et al., 2017; Florensa et al., 2017) attempt to split the goal declaration and goal achievement across different stages and layers of the learned architecture. Thus, while hierarchical construction of goals follows a set of coded rules, our action grammars are inferred entirely data-driven based on agent experience. Usually, the top level of the hierarchy specifies goals in the environment while the lower levels have to achieve them. Again, such architectures lack sample efficiency and easy interpretation. Our context-free grammar-based approach, on the other hand, is a symbolic method that requires few rollout traces and generalizes to more difficult task-settings.

Finally, unlike recent work on unifying symbolic and connectionist methods, we do not aim to discover relationships between entities (Garnelo et al., 2016; Zambaldi et al., 2018). Instead our proposed action grammar framework achieves interpretability by extracting hierarchical subroutines associated with sub-goal achievements (Beyret et al., 2019).

## 3 METHODOLOGY

The Action Grammar Reinforcement Learning framework operates by having the agent repeatedly iterate through two steps (as laid out in Figure 1):

(A) **Gather Experience**: the base off-policy RL agent interacts with the environment and stores its experiences

(B) **Identify Action Grammar**: the experiences are used to identify the agent's action grammar which is then appended to the agent's action set in the form of macro-actions

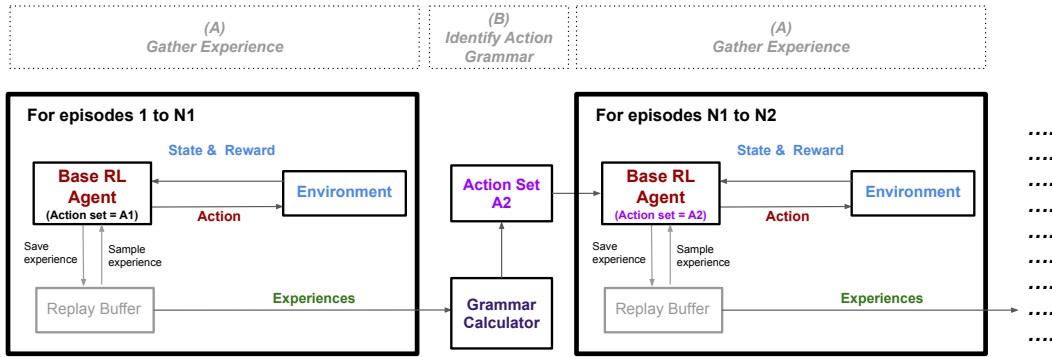

**Figure 1:** The high-level steps involved in the AG-RL algorithm. In the first Gather Experience step the base agent interacts as normal with the environment for N1 episodes. Then we feed the experiences of the agent to a grammar calculator which outputs macro-actions that get appended to the action set. The agent starts interacting with the environment again with this updated action set. This process repeats as many times as required, as set through hyperparameters.

During the first **Gather Experience** step of the game the base RL agent plays normally in the environment for some set number of episodes. The only difference is that during this time we occasionally run an episode of the game with all random exploration turned off and store these experiences separately. We do this because we will later use these experiences to identify the action grammar and so we do not want them to be influenced by noise. See left part of Figure 1.

After some set number of episodes we pause the agent's interaction with the environment and enter the first **Identify Action Grammar** stage, see middle part of Figure 1. This firstly involves collecting the actions used in the best performing of the no-exploration episodes mentioned above. We then feed these actions into a grammar calculator which identifies the action grammar. A simple choice for the grammar calculator is Sequitur (Nevill-Manning & Witten, 1997). Sequitur receives a sequence of actions as input and then iteratively creates new symbols to replace any repeating sub-sequences of actions. These newly created symbols then represent the macro-actions of the action grammar. To minimise the influence of noise on grammar generation, however, we need to regularise the process. A naive regulariser is k-Sequitur (Stout et al., 2018) which is a version of Sequitur that only creates a new symbol if a sub-sequence repeats at least $k$ times (instead of at least two times), where higher $k$ corresponds to stronger regularisation. Here we use a more principled approach and regularise on the basis of an information theoretic criterion: we generate a new symbol if doing so reduces the total amount of information needed to encode the sequence.

After we have identified the action grammar we enter our second **Gather Experience** step. This firstly involves appending the macro-actions in the action grammar to the agent's action set. To do this without destroying what our q-network and/or policy has already learned we use transfer learning. For every new macro-action we add a new node to the final layer of the network, leaving all other nodes and weights unchanged. We also initialise the weights of each new node to the weights of their first primitive action (as it is this action that is most likely to have a similar action value to the macro-action). e.g. if the primitive actions are $\{a, b\}$ and we are adding macro-action *abb* then we initialise the weights of the new macro-action to those of $a$.

Then our agent begins interacting with the environment as normal but with an action set that now includes macro-actions and four additional changes:

i) In order to maximise information efficiency, when storing experiences from this stage onwards we use a new technique we call Hindsight Action Replay (HAR). It is related to Hindsight Experience Replay which creates new experiences by reimagining the goals the agent was trying to achieve. Instead of reimagining the goals, HAR creates new experiences by reimagining the *actions*. In particular it reimagines them in two ways:

1. If we play a macro-action then we also store the experiences as if we had played the sequence of primitive actions individually

2. If we play a sequence of primitive actions that matches an existing macro-action then we also store the experiences as if we had played the macro-action

See Appendix A for an example of how HAR is able to more than double the number of collected experiences in some cases.

ii) To make sure that the longer macro-actions receive enough attention while learning, we sample experiences from an action balanced replay buffer. This acts as a normal replay buffer except it returns samples of experiences containing equal amounts of each action.

iii) To reduce the variance involved in using long macro-actions we develop a new technique called Abandon Ship. During every timestep of conducting a macro-action we calculate how much worse it is for the agent to continue executing its macro-action compared to abandoning the macro-action and picking the highest value primitive action instead. Formally we calculate this value as $d = 1 - \frac{\exp(q_m)}{\exp(q_{highest})}$ where $q_m$ is the action value of the primitive action we are conducting as part of the macro-action and $q_{highest}$ is the action value of the primitive action with the highest action value. We also store the moving average, $m(d)$, and moving standard deviation, $std(d)$, of $d$. Then each timestep we compare $d$ to threshold $t = m(d) + std(d)z$ where $z$ is the abandon ship hyperparameter that determines how often we will abandon macro-actions. If $d > t$ then we abandon our macro-action and return control back to the policy, otherwise we continue executing the macro-action.

---

**Algorithm 1** AG-RL

---

1: Initialise environment env, base RL algorithm R, replay buffer D and action set A
2: **for** each iteration **do**
3:     $F \leftarrow$ GATHER_EXPERIENCE$(A)$
4:     $A \leftarrow$ IDENTIFY_ACTION_GRAMMAR$(F)$
5:
6: **procedure** GATHER_EXPERIENCE(A)
7:     transfer_learning(A)         ▷ If action set changed do transfer learning
8:     $F \leftarrow \emptyset$     ▷ Initialise F to store no-exploration episode experiences
9:     **for** each episode **do**
10:         **if** no exploration time **then** turn off exploration     ▷ Periodically turn off exploration
11:         $E \leftarrow \emptyset$     ▷ Initialise E to store an episode's experiences
12:         **while** not done **do**
13:             $ma_t = $R.pick_action$(s_t)$     ▷ Pick next primitive action / macro-action
14:             **for** $a_t$ in $ma_t$ **do**     ▷ Iterate through each primitive action in the macro-action
15:                 **if** abandon_ship$(s_t, a_t)$ **then** break     ▷ Abandon macro-action if required
16:                 $s_{t+1}, r_{t+1}, d_{t+1} = $env.step$(a_t)$     ▷ Play action in environment
17:                 $E \leftarrow E \cup \{(s_t, a_t, r_{t+1}, s_{t+1}, d_{t+1})\}$     ▷ Store the episode's experiences
18:                 R.learn(D)     ▷ Learning iteration for base RL algorithm
19:         $D \leftarrow D \cup$ HAR$(E)$     ▷ Use HAR when updating replay buffer
20:         **if** no exploration time **then** $F \leftarrow F \cup E$     ▷ Store no-exploration experiences
21:     **return** F
22:
23: **procedure** IDENTIFY_ACTION_GRAMMAR(F)
24:     $F \leftarrow$ extract_best_episodes(F)     ▷ Keep only the best performing no-exploration episodes
25:     action_grammar $\leftarrow$ grammar_algorithm(F)     ▷ Infer action grammar using experiences
26:     $A \leftarrow A \cup$ action_grammar     ▷ Update the action set with identified macro-actions
27:     **return** A

---

iv) When our agent is picking random exploration moves we bias its choices towards macro-actions. For example, when a DQN agent picks a random move (which it does epsilon proportion of the time) we set the probability that it will pick a macro-action, rather than a primitive action, to the higher probability given by the hyperparameter "Macro Action Exploration Bonus". In these cases, we do not use Abandon Ship and instead let the macro-actions fully roll out.

The second Gather Experience step then continues until it is time to do another Identify Action Grammar step or until the agent has been trained for long enough and the game ends. Algorithm 1 provides the full AG-RL algorithm.

## 4   SIMPLE EXAMPLE

We now highlight the core aspects of how AG-RL works using the simple game Towers of Hanoi. The game starts with a set of disks placed on a rod in decreasing size order. The objective of the game is to move the entire stack to another rod while obeying the following rules: i) Only one disk can be moved at a time; ii) Each move consists of taking the upper disk from one of the stacks and placing it on top of another stack or on an empty rod; and iii) No larger disk may be placed on top of a smaller disk. The agent only receives a reward when the game is solved, meaning rewards are sparse and that it is difficult for an RL agent to learn the solution. Figure 2 runs through an example of how the game can be solved with the letters '$a$' to '$f$' being used to represent the 6 possible moves in the game.

In this game, AG-RL proceeds by having the base agent (which can be any off-policy RL agent) play the game as normal. After some period of time we pause the agent and collect some of the actions taken by the agent, e.g. say the agent played the sequence of actions: "*bafbcdbafecfbaf-bcdbcfecdbafbcdb*". Then we use a grammar induction algorithm such as Sequitur to create new symbols to represent repeating sub-sequences. In this example, Sequitur would create the 4 new symbols: $\{G : bc, H : ec, I : baf, J : bafbcd\}$. We then append these symbols to the agent's action set as macro-actions, so that the action set goes from $A = \{a, b, c, d, e, f\}$ to:

$$A = \{a, b, c, d, e, f\} \cup \{bc, ec, baf, bafbcd\}$$

The agent then continues playing in the environment with this new action set which includes macro-actions. Because the macro-actions are of length greater than one it means that their usage effectively reduces the time dimensionality of the problem, making it an easier problem to solve in some cases.[1]

We now demonstrate the ability of AG-RL to consistently improve sample efficiency on the much more complicated Atari suite setting.

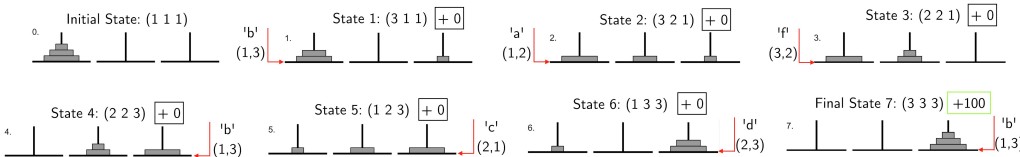

**Figure 2:** Example of a solution to the Towers of Hanoi game. Letters $a$ to $f$ represent the 6 different possible moves. After 7 moves the agent has solved the game and receives +100 reward.

## 5   RESULTS

We first evaluate the AG-RL framework using DDQN as the base RL algorithm. We refer to this as AG-DDQN. We compare the performance of AG-DDQN and DDQN after training for 350,000 steps on 8 Atari games chosen a priori to represent a broad range. To accelerate training times for both we set their convolutional layer weights as equal to those of some pre-trained agents[2] and then only train the fully connected layers.

---

[1]Experimental results for this simplified setting may be found in section F of the appendix.

[2]We used the pre-trained agents in the GitHub repository rl-baselines-zoo https://github.com/araffin/rl-baselines-zoo/tree/master/trained_agents/dqn

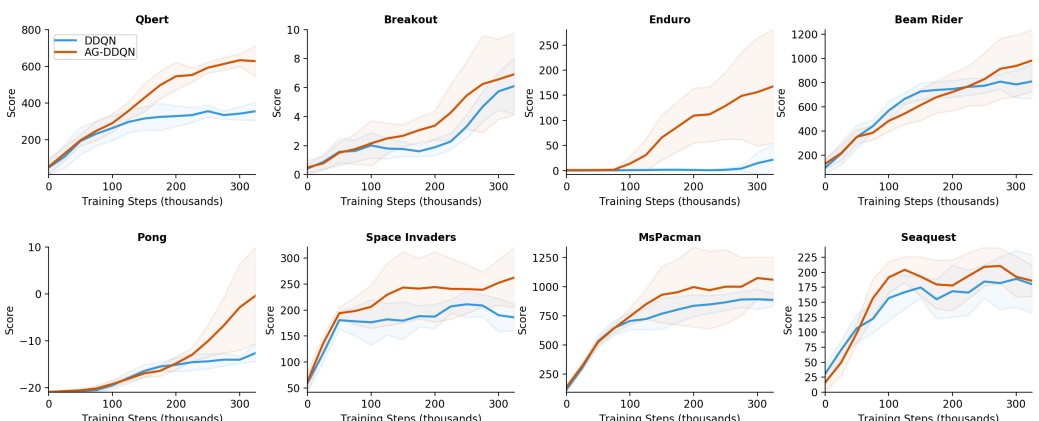

**Figure 3:** Comparing AG-DDQN to DDQN for 8 Atari games. Graphs show the average evaluation score over 5 random seeds where an evaluation score is calculated every 25,000 steps and averaged over the previous 3 scores. For the evaluation methodology we used the same no-ops condition as in van Hasselt et al. (2015). The shaded area shows ±1 standard deviation.

For the DDQN-specific hyperparameters of both networks we do no hyperparameter tuning and instead use the hyperparmaeters from van Hasselt et al. (2015) or set them manually. For the AG-specific hyperparameters we tried four options for the abandon ship hyperparameter (No Abandon Ship, 1, 2, 3) for the game Qbert and chose the option with the highest score. No other hyperparameters were tuned and all games then used the same hyperparameters which can all be found in Appendix B along with a more detailed description of the experimental setup.

We find that *AG-DDQN outperforms DDQN in all 8 games with a median final improvement of 31% and a maximum final improvement of 668%* - Figure 3 summarises the results.

Next we further evaluate AG-RL by using SAC as the base RL algorithm, leading to AG-SAC. We compare the performance of AG-SAC to SAC. We train both algorithms for 100,000 steps on 20 Atari games chosen a priori to represent a broad range games. As SAC is a much more efficient algorithm than DDQN this time we train both agents from scratch and do not use pre-trained convolutional layers.

For the SAC-specific hyperparameters of both networks we did no hyperparameter tuning and instead used a mixture of the hyperparameters found in Haarnoja et al. (2018) and Kaiser et al. (2019). For the AG-specific hyperparmaeters again the only tuning we did was amongst 4 options for the abandon ship hyperparameter (No Abandon Ship, 1, 2, 3) on the game Qbert. No other hyperparameters were tuned and all games then used the same hyperparameters, details of which can be found in Appendix C.

Our results show *AG-SAC outperforms SAC in 19 out of 20 games with a median improvement of 96% and a maximum improvement of 3,756%* - Figure 4 summarises the results and Appendix D provides them in more detail.

We also find that *AG-SAC outperforms Rainbow, which is the model-free state-of-the-art for Atari sample efficiency, in 17 out of 20 games with a median improvement of 62% and maximum improvement of 13,140%* - see Appendix D for more details. Also note that the Rainbow scores used were taken from Kaiser et al. (2019) who explain they were the result of extensive hyperparameter tuning compared to our AG-SAC scores which benefited from very little hyperparameter tuning.

## 6 DISCUSSION

To better understand the results, we first explore what types of macro-actions get identified during the Identify Action Grammar stage, whether the agents use them extensively or not, and to what extent the Abandon Ship technique plays a role. We find that the length of macro-actions can vary greatly from length 2 to over 100. An example of an inferred macro-action was *8888111188881111*

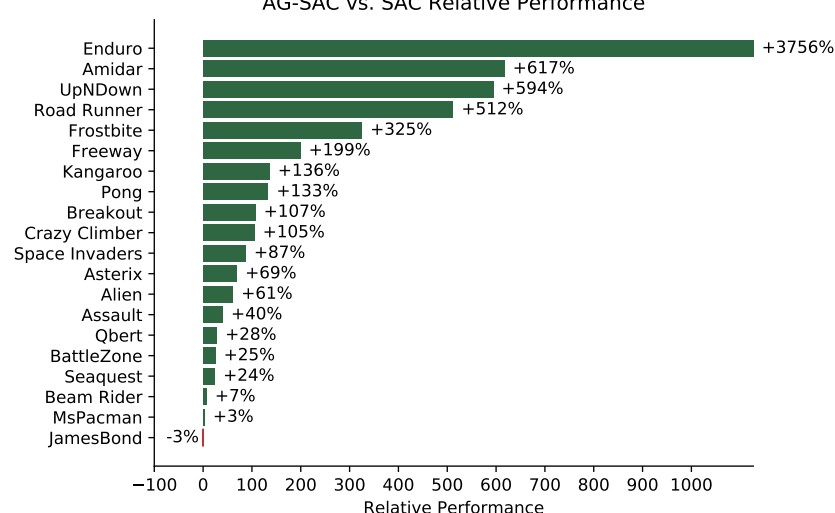

**Figure 4:** Comparing AG-SAC to SAC for 20 Atari games. Graphs show the average evaluation score over 5 random seeds where an evaluation score is calculated at the end of 100,000 steps of training. For the evaluation methodology we used the same no-ops condition as in van Hasselt et al. (2015).

from the game Beam Rider where 1 represents Shoot and 8 represents move Down-Right. This macro-action seems particularly useful in this game as the game is about shooting enemies whilst avoiding being hit by them.

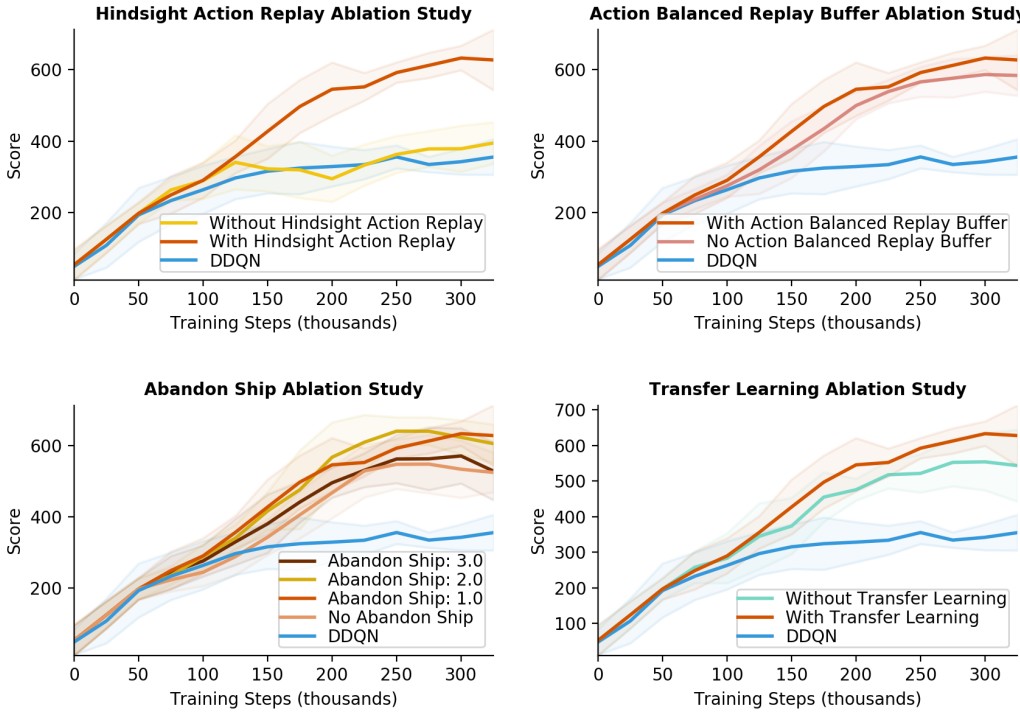

**Figure 5:** Ablation study comparing DDQN to different versions of AG-DDQN for the game Qbert. The dark line is the average of 5 random seeds, the shaded area shows ±1 standard deviation across the seeds.

We also found that the agents made extensive use of the macro-actions. Taking each game's best performing AG-SAC agent, the average *attempted* move length during evaluation was 20.0. Because

of Abandon Ship the average *executed* move length was significantly lower at 6.9 but still far above the average of 1.0 we would get if the agents were not using their macro-actions. Appendix E gives more details on the differences in move lengths between games.

We now conduct an ablation study to investigate the main drivers of AG-DDQN's performance in the game Qbert, with results in Figure 5 . Firstly we find that HAR was crucial for the improved performance and without it AG-DDQN performed no better than DDQN. We suspect that this is because without HAR there are much fewer experiences to learn from and so our action value estimates have very high variance.

Next we find that using an action balanced replay buffer improved performance somewhat but by a much smaller and potentially insignificant amount. This potentially implies that it may not be necessary to use an action balanced replay and that the technique may work with an ordinary replay buffer. We also see that with our chosen abandon ship hyperparameter of 1.0, performance was higher than when abandon ship was not used. Performance was also similar for the choice of 2.0 which suggests performance was not too sensitive to this choice of hyperparameter. Finally we see improved performance from using transfer learning when appending macro-actions to the agent's action set rather than creating a new network.

Lastly, we note that the game in which AG-DDQN and AG-SAC both do relatively best in is Enduro. Enduro is a game with sparse rewards and therefore one where exploration is very important. We therefore speculate that AG-RL does best on this game because using long macro-actions increases the variance of where an agent can end up and therefore helps exploration.

## 7  CONCLUSION

Motivated by the parallels between the hierarchical composition of language and that of actions, we combine techniques from computational linguistics and RL to help develop the Action Grammars Reinforcement Learning framework.

The framework expands on two key areas of RL research: Symbolic RL and Hierarchical RL. We extend the ideas of symbolic manipulation in RL (Garnelo et al., 2016; Garnelo & Shanahan, 2019) to the dynamics of sequential action execution. Moreover, while Relational RL approaches (Zambaldi et al., 2018) draw on the complex logic-based framework of inductive programming, we merely observe successful behavioral sequences to induce higher order structures.

We provided two implementations of the framework: AG-DDQN and AG-SAC. We showed that AG-DDQN improves on DDQN in 8 out of 8 tested Atari games (median +31%, max +668%) and AG-SAC improves on SAC in 19 out of 20 tested Atari games (median +96%, max +3,756%) all without substantive hyperparameter tuning. We also show that AG-SAC beats the model-free state-of-the-art for 17 out of 20 Atari games (median +62%, max +13,140%) in terms of sample efficiency, again even without substantive hyperparameter tuning.

As part of AG-RL we also provided two new and generally applicable techniques: Hindsight Action Replay and Abandon Ship. Hindsight Action Replay can be used to drastically improve information efficiency in any off-policy setting involving macro-actions. Abandon Ship reduces the variance involved when training macro-actions, making it feasible to train algorithms with very long macro-actions (over 100 steps in some cases).

Overall, we have demonstrated the power of action grammars to consistently improve the performance of RL agents. We believe our work is just one of many possible ways of incorporating the concept of action grammars into RL and we look forward to exploring other methods.

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

APPENDIX

## A  HINDSIGHT ACTION REPLAY

Below we provide an example of how HAR stores the experiences of an agent after they played the moves *acab* where *a* and *b* are primitive actions and *c* represents the macro-action *ababa*.

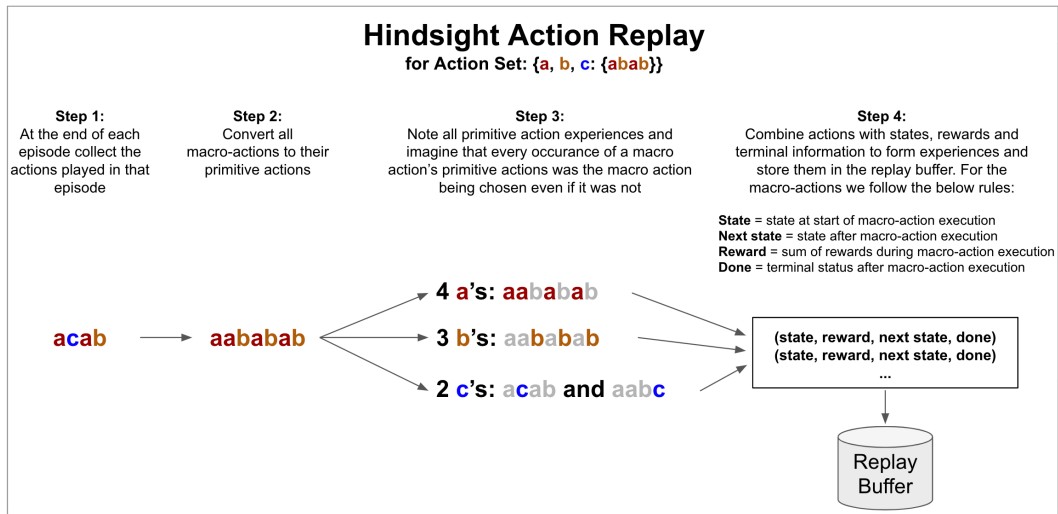

**Figure 6:** The Hindsight Action Replay (HAR) process with Action Set: $\{a, b, c:\{abab\}\}$ meaning that there are 2 primitive actions *a* and *b*, and one macro-action *c* which represents the sequence of primitive actions *abab*. The example shows how using HAR leads to the original sequence of 4 actions *acab* producing 9 experiences for the replay buffer instead of only 4.

## B  AG-DDQN Experiment and Hyperparameters

The hyperparameters used for the DDQN results are given by Table 1. The network architecture was the same as in the original Deepmind Atari paper (Mnih et al., 2015).

**Table 1:** Hyperparameters used for DDQN and AG-DDQN results

| Hyperparameter | Value |
|---|---|
| Batch size | 32 |
| Replay buffer size | 1,000,000 |
| Discount rate | 0.99 |
| Steps per learning update | 4 |
| Learning iterations per round | 1 |
| Learning rate | 0.0005 |
| Optimizer | Adam |
| Weight Initialiser | He |
| Min Epsilon | 0.1 |
| Epsilon decay steps | 1,000,000 |
| Fixed network update frequency | 10000 |
| Loss | Huber |
| Clip rewards | Clip to [-1, +1] |
| Initial random steps | 25,000 |

The architecture and hyperparameters used for the AG-DDQN results that are relevant to DDQN are the same as for DDQN and then the rest of the hyperparameters are given by Table 2.

**Table 2:** Hyperparameters used for AG-DDQN results

| Hyperparameter | Value | Description |
|---|---|---|
| Evaluation episodes | 5 | The number of no exploration episodes we use to infer the action grammar |
| Replay buffer type | Action balanced | The type of replay buffer we use |
| Steps before inferring grammar | 75,001 | The number of steps we run before inferring the action grammar |
| Abandon ship | 1.0 | The threshold for the abandon ship algorithm |
| Macro-action exploration bonus | 4.0 | How much more likely we are to pick a macro-action than a primitive action when acting randomly |

## C    AG-SAC HYPERPARAMETERS

The hyperparameters used for the discrete SAC results are given by Table 3. The network architecture for both the actor and the critic was the same as in the original Deepmind Atari paper (Mnih et al., 2015).

**Table 3:** Hyperparameters used for SAC and AG-SAC results

| Hyperparameter | Value |
|---|---|
| Batch size | 64 |
| Replay buffer size | 1,000,000 |
| Discount rate | 0.99 |
| Steps per learning update | 4 |
| Learning iterations per round | 1 |
| Learning rate | 0.0003 |
| Optimizer | Adam |
| Weight Initialiser | He |
| Fixed network update frequency | 8000 |
| Loss | Mean squared error |
| Clip rewards | Clip to [-1, +1] |
| Initial random steps | 20,000 |

The architecture and hyperparameters used for the AG-SAC results that are relevant to SAC are the same as for SAC and then the rest of the hyperparameters are given by Table 4.

**Table 4:** Hyperparameters used for AG-SAC results

| Hyperparameter | Value | Description |
|---|---|---|
| Evaluation episodes | 5 | The number of no exploration episodes we use to infer the action grammar |
| Replay buffer type | Action balanced | The type of replay buffer we use |
| Steps before inferring grammar | 30,001 | The number of steps we run before inferring the action grammar |
| Abandon ship | 2.0 | The threshold for the abandon ship algorithm |
| Macro-action exploration bonus | 4.0 | How much more likely we are to pick a macro-action than a primitive action when acting randomly |
| Post inference random steps | 5,000 | The number of random steps we run immediately after updating the action set with new macro-actions |

# D   SAC, AG-SAC AND RAINBOW ATARI RESULTS

Below we provide all SAC, AG-SAC and Rainbow results after running 100,000 iterations for 5 random seeds. We see that AG-SAC improves over SAC in 19 out of 20 games (median improvement of 96%, maximum improvement of 3756%) and that AG-SAC improves over Rainbow in 17 out of 20 games (median improvement 62%, maximum improvement 13140%).

**Table 5:** SAC, AG-SAC and Rainbow results on 20 Atari games. The mean result of 5 random seeds is shown with the standard deviation in brackets. As a benchmark we also provide a column indicating the score an agent would get if it acted purely randomly.

| Game | Random | Rainbow | SAC | AG-SAC |
|---|---|---|---|---|
| Enduro | 0.0 | 0.53 | 0.8 (0.8) | 30.1 (10.1) |
| Amidar | 11.8 | 20.8 | 7.9 (5.1) | 56.7 (15.4) |
| UpNDown | 488.4 | 1346.3 | 250.7 (176.5) | 1739.1 (835.8) |
| Road Runner | 0.0 | 524.1 | 305.3 (557.4) | 1868.7 (1658.3) |
| Frostbite | 74.0 | 140.1 | 59.4 (16.3) | 252.3 (79.8) |
| Freeway | 0.0 | 0.1 | 4.4 (9.9) | 13.2 (12.1) |
| Kangaroo | 42.0 | 38.7 | 29.3 (55.1) | 69.3 (80.4) |
| Pong | -20.4 | -19.5 | $-20.98$ (0.0) | $-20.95$ (0.1) |
| Breakout | 0.9 | 3.3 | 0.7 (0.5) | 1.5 (1.4) |
| Crazy Climber | 7339.5 | 12558.3 | 3668.7 (600.8) | 7510.0 (3898.7) |
| Space Invaders | 148.0 | 135.1 | 160.8 (17.3) | 301.0 (75.1) |
| Asterix | 248.8 | 285.7 | 272.0 (33.3) | 459.0 (104.2) |
| Alien | 184.8 | 290.6 | 216.9 (43.0) | 349.7 (33.4) |
| Assault | 233.7 | 300.3 | 350.0 (40.0) | 490.6 (119.0) |
| Qbert | 166.1 | 235.6 | 280.5 (124.9) | 359.7 (172.8) |
| BattleZone | 2895.0 | 3363.5 | 4386.7 (1163.0) | 5486.7 (1461.7) |
| Seaquest | 61.1 | 206.3 | 211.6 (59.1) | 261.9 (56.3) |
| Beam Rider | 372.1 | 365.6 | 432.1 (44.0) | 463.0 (219.1) |
| MsPacman | 235.2 | 364.3 | 690.9 (141.8) | 712.9 (194.5) |
| JamesBond | 29.2 | 61.7 | 68.3 (26.2) | 66.3 (19.4) |

Note that the scores for Pong are negative and so to calculate the proportional improvement for this game we first convert the scores to their increment over the minimum possible score. In Pong the minimum score is -21.0 and so we first add 21 to all scores before calculating relative performance.

Also note that for Pong both AG-SAC and SAC perform worse than random. The improvement of AG-SAC over SAC for Pong therefore could be considered a somewhat spurious result and potentially should be ignored. Note that there are no other games where AG-SAC performs worse than random though and so this issue is contained to the game Pong.

# E   MACRO-ACTIONS AND ABANDON SHIP

**Table 6:** The lengths of moves attempted and executed in the best performing AG-SAC seed for each game. Executed move lengths being shorter than attempted move lengths indicates that the Abandon Ship technique was used. The games are ordered in terms of the percentage improvement AG-SAC saw over SAC with the first game being the game that saw the most improvement.

| Game | Attempted Move Lengths | Executed Move Lengths | Executed / Attempted |
|---|---|---|---|
| Enduro | 14.8 | 12.0 | 81.1% |
| Amidar | 3.4 | 2.8 | 82.4% |
| UpNDown | 16.4 | 11.6 | 71.0% |
| Road Runner | 8.2 | 6.8 | 82.9% |
| Frostbite | 2.0 | 2.0 | 100.0% |
| Freeway | 15.8 | 15.2 | 96.2% |
| Kangaroo | 2.1 | 1.5 | 73.8% |
| Breakout | 7.9 | 2.2 | 27.8% |
| Crazy Climber | 7.1 | 4.4 | 62.0% |
| Space Invaders | 8.7 | 6.3 | 72.4% |
| Asterix | 58.3 | 9.8 | 16.8% |
| Alien | 13.1 | 11.6 | 88.5% |
| Assault | 126.9 | 8.1 | 6.4% |
| Qbert | 8.2 | 8.0 | 97.6% |
| Battle Zone | 63.4 | 6.0 | 9.5% |
| Seaquest | 14.0 | 8.6 | 61.4% |
| Beam Rider | 12.2 | 5.8 | 47.5% |
| MsPacman | 8.2 | 7.5 | 91.5% |
| Pong | 8.4 | 6.5 | 77.4% |
| James Bond | 1.3 | 1.2 | 97.8% |

## F  ADDITIONAL EXPERIMENTS

The Towers of Hanoi experiments depicted in the figure below are run with SMDP-Q-Learning. Let $r_{\tau_m} = \sum_{i=1}^{\tau_m} \gamma^{i-1} r_{t+i}$ denote the accumulated and discounted reward for executing a macro. Tabular value estimates can then be updated using SMDP-Q-Learning (Bradtke & Duff, 1995; Parr, 1998) in a model-free bootstrapping-based manner:

$$Q(s,m)_{k+1} = (1-\alpha)Q(s,m)_k + \alpha\left( r_{\tau_m} + \gamma^{\tau_m} \max_{m' \in \mathcal{M}} Q(s',m')_k \right) \qquad (1)$$

We do not make use of HAR or "Abandon Ship" in these experiments and use the following hyperparameters:

| Tabular Action Grammar SMDP-Q-Learning Hyperparameters: | | | |
|---|---|---|---|
| Hyperparameter | Value | Hyperparameter | Value |
| Learning rate $\alpha$ | 0.8 | Discount factor $\gamma$ | 0.95 |
| Eligibility Trace $\lambda$ | 0 | Exploration $\epsilon$ | 0.1 |

The $TD(\lambda)$ baseline shares all the hyperparameters apart from the eligibility trace $\lambda$ which is set to 0.1. We train the agents 300000 (5 disks) and 7000000 (6 disks) steps.

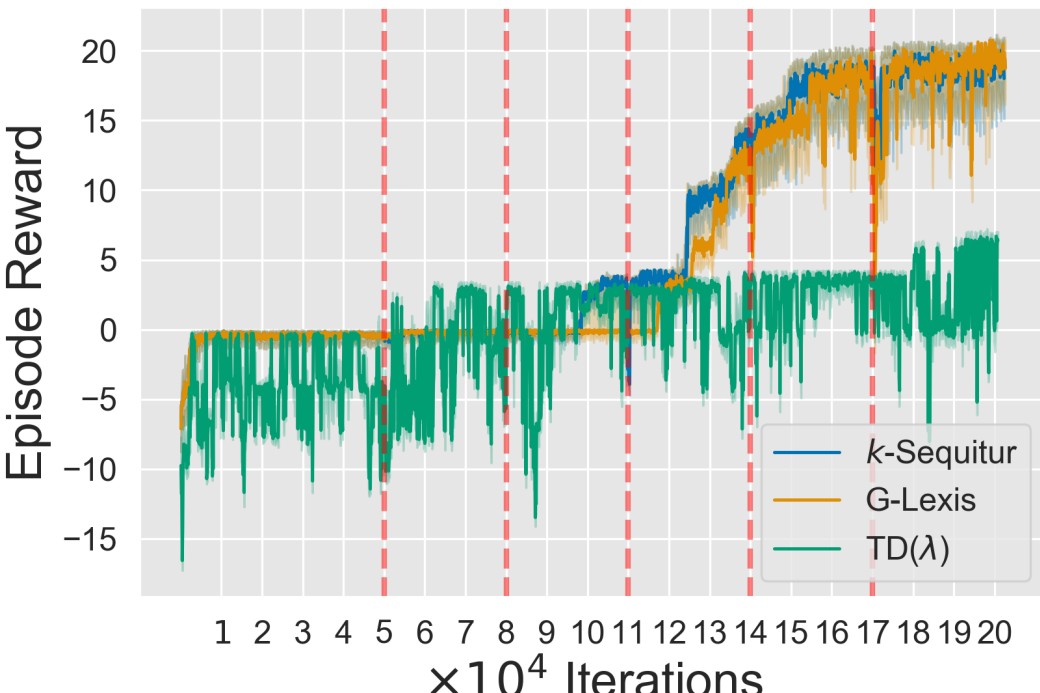

**Figure 7:** Action Grammar Reinforcement Learning for Towers of Hanoi (5 Disk Environment). Red vertical lines correspond to grammar updates.

