# OpenReview forum: "Reinforcement Learning with Structured Hierarchical Grammar Representations of Actions"
_ICLR.cc/2020/Conference — Reject_

### Official Review · AnonReviewer1 · 2019-10-23
**Official Blind Review #1**

**Rating:** 3

**Review:**

The authors propose a method for learning macro-actions in a multi-step manner, where Sequitur, a grammar calculator, is leveraged together with an entropy-minimisation based strategy to find relevant macro-actions. The authors propose a system to bootstrap the weights of these macro-actions when increasing the policy's action space, and a system to increase the amount of data (and bias it towards macro-actions) used to learn a policy for when conditioned on this increased action-space. The authors test against a subset of the Arcade Learning Environment suite.

Overall, I'm conflicted by this paper. On one hand, the framework is interesting, and their method involves the usage and exploration of quite a few nice ideas; on the other hand, (a) the quality of the scientific contribution is hard to judge considering the significant differences between the proposed baselines and and their methods, and (b) the experimental section doesn't provide a lot of qualitative analysis and signal wrt. each component.

Furthermore, I have the following issues / questions:

1. I'm not convinced that the usage of Sequitur to build the macro-actions is sufficient to declare this work novel wrt. other macro-action papers. Sequitur usage in this case seems to be particularly overkill, since ultimately all that the method seems to be doing is finding frequent sequences of actions, which can be done quite fast (at least given the amount of training steps) simply using search and pattern matching. From my point of view, there doesn't seem to be a lot in that work that exploits the fact that the macro-actions are constructed as a "grammar" (beyond, maybe, HAR)

2. The Abandon Ship heuristics is effectively a fixed termination policy, which makes the entire setup somewhat similar to options. In this case, what is traded is learning complexity for a hyperparameter and a significant restriction in how the macro-actions terminate. Did you attempt to learn this function at all? Do you have any insights / experiments that might show how the heuristics behaves with changing values of $z$? Would it be possible to plot the distribution of attempted vs executed move lengths rather than then averages (since I doubt they would be normally distributed)?

3. Given points 1 and 2, the literature review is lacking - there's a lot of prior work done on macro-actions in both RL and robotics (planning, HRI, ...) that goes well beyond the few recent papers mentioned by the authors, and I think it might be necessary to mention work on options where the termination function is structured / biased in some way.

4. I have some doubt the experimental setup for DDQN fairly gives a fair assessment of the method. When using a pretrained features, the problem becomes significantly easier, and thus AG-DDQN potentially doesn't need to deal with the problem of learning extremely bad / noisy macro-actions. I would love to see the method trained for a more reasonable amount of frames without pre-training. Also, did the 8 / 20 atari games get chosen randomly, or were they picked based on some environment features?

5. How do the Q-values for the policy evolve with training time? The proposed methods seem to somewhat imply that the action space grows unboundedly, which might seriously destroy the policy for tasks that require much longer training. Would it be possible to add a paragraph about how the policies evolve in at least some of these environments? Are macro-actions used most of the times after some full iterations? How many <learning -> action distillation> iterations are actually done in the current experiments?

At this point, I cannot recommend the acceptance of this work, however I'd be willing to reconsider my rating if the authors address the above points.


**Experience Assessment:**

I have published in this field for several years.

**Review Assessment: Checking Correctness Of Derivations And Theory:**

I carefully checked the derivations and theory.

**Review Assessment: Checking Correctness Of Experiments:**

I carefully checked the experiments.

**Review Assessment: Thoroughness In Paper Reading:**

I read the paper thoroughly.

---

> ### Author Response · Authors · 2019-11-13
> **Rebuttal with brief description of revised submission**
>
> Dear reviewer 1,
>
> We are very thankful for your comments and believe that multiple issues of importance are being raised.
>
> Regarding point 1: The k-Sequitur algorithm runs in linear time in the length of the presented action sequence. Hence, in computational terms it is easily feasible. Furthermore, the entropy regularisation deployed in the technique makes it more than a greedy compression technique. Instead, the main point that we want to raise is that the grammatical inference procedure obtains a hierarchical representation of actions. A key advantage of this symbolic procedure is the interpretability of such representations. For now, we leave this for future work.
>
> Regarding point 2: The relationship between abandon ship and termination policies is a very interesting observation. We have not attempted to learn the termination in an end-to-end fashion. Our current understanding is that this poses significant challenges to options (see concurrent work by Harutyunyan et al., 2019  https://arxiv.org/pdf/1902.09996.pdf) and it is not entirely trivial how to combat this additional non-stationary component. For now, the simple moving average based heuristic has sufficed and reduces the complexity of the proposed algorithm.
>
> Regarding point 3: We agree and have updated the manuscript to include a more detailed literature review, see section 2 of the revised paper.
>
> Regarding point 4: Yes, we agree. It is easier to infer effective macro-actions based on already successful on-policy rollouts. We want to highlight that this provides a potential future research direction, i.e. skill distillation/imitation learning via action grammar inference. Furthermore and to address your point, the results of the Action Grammar SAC agents are obtained without pre-training. And again, the agents do experience a significant speed up  in learning after the first grammar is inferred (see figure 4, performance after 100,000 transitions). Finally, as already stated we have experimented with a tabular example in Towers of Hanoi where grammar macro-actions are also without pre-training - see new appendix item F.
>
> Best wishes,
> The authors.

---

> > ### Comment · AnonReviewer1 · 2019-11-14
> > **Response to rebuttal**
> >
> > >The k-Sequitur algorithm runs in linear time in the length of the presented action sequence. Hence, in computational terms it is easily feasible. Furthermore, the entropy regularisation deployed in the technique makes it more than a greedy compression technique.
> >
> > I think it's fairly clear that k-Sequitur does more than greedy compression, however my point was that I don't see a discussion about what this additional complexity buys to the policy learning process, and what the tradeoffs are of using Sequitur rather than - say - greedy search.
> >
> >
> > >Instead, the main point that we want to raise is that the grammatical inference procedure obtains a hierarchical representation of actions. A key advantage of this symbolic procedure is the interpretability of such representations. For now, we leave this for future work.
> >
> > Right, but if this "key advantage" is not exploited (as far as I can see), then it is not an advantage at all, at least wrt this particular publication.
> >
> > Think about this issue from the perspective of someone that needs to build on your work: what is the "simplest" combination - of the ideas you have introduced - that shows the properties you have demonstrated through this method? What is the _scientific knowledge_ that one gains from reading your paper?
> >
> >
> > >the simple moving average based heuristic has sufficed and reduces the complexity of the proposed algorithm.
> >
> > I think even just the fact that learning termination functions is a common HRL problem tells me that it is fundamentally important to deal with multi-stage policies, and it's unwise to present "abandon ship" without comparing it to previous work in the area.
> >
> > However, ultimately my main concern is that the heuristic is just that, a heuristic: it's bound to have corner cases and fail to generalise to interesting settings, and a proper evaluation of the system would include a discussion on failure cases and unexpected behaviour, which I don't really see in the manuscript?
> >
> >
> > >We agree and have updated the manuscript to include a more detailed literature review, see section 2 of the revised paper.
> >
> > Thank you for that, it looks better.
> >
> >
> > >Yes, we agree. It is easier to infer effective macro-actions based on already successful on-policy rollouts.
> >
> > Would it be possible to add any experiment / analysis showing the degree of how much this matters?
> >
> >
> > >And again, the agents do experience a significant speed up  in learning after the first grammar is inferred (see figure 4, performance after 100,000 transitions).
> >
> > Right, but *why* is that the case? Does it mean that the policies are just facilitated in exploration? Do the initial few macros still retain usefulness towards the end of the training stage? What is the evolution of the distribution in terms of action usage across these tasks?
> >
> > Sample complexity is a poor way of analysing this sort of methods, since it's difficult to disentangle behaviour caused by task settings rather than properties of the methods, so the analysis would be better if it were to be augmented with some qualitative, method-specific, data.

---

### Official Review · AnonReviewer3 · 2019-10-23
**Official Blind Review #3**

**Rating:** 8

**Review:**

This paper introduced a way to combine actions into meta-actions through action grammar. The authors trained agents that executes both primitive actions and meta-actions, resulting in better performance on Atari games. Specifically, meta-actions are generated after a period of training from collected greedy action sequences by finding repeated sub-sequences of actions. Several tricks are used to speed up learning and to make the framework more flexible. The most effective one is HAR (hindsight action replay), without which the agent's performance reduces to that of the baseline.

Overall, this paper could be a great contribution for the following reasons:
1. The paper is well written, with clear performance advantages over the baseline.
2. The paper provides a different perspective for HRL research, namely that we might not need to have a hierarchical policy to benefit from hierarchical actions that spans over many timesteps.
3. From this paper's ablation study for HAR, it seems to suggest that even with similar experiences, one can get better performance by substituting actions with temporally abstracted actions, propagating value function errors further back in time. If so, this work can serve as a novel counterexample to the claim made in Nachum et al., 2019.

The authors may want to address the following:
1. They may want to compare and contrast to other works in HRL that also does temporally abstracted actions. e.g. h-DQN, Feudal networks. Or even to repeating the same action N times-- a simple trick commonly used in Atari  -- which can be seen as a very naive form of action grammar.
2. The main claim that having Action Grammar improves sample efficiency is not proved clearly. Apart from the ablation study, it's not immediately clear whether sticking to sub-sequence of actions are inherently beneficial for exploration, or that the agent somehow learned faster with the same set of samples collected.
3. It seems that the algorithm may be the most effective in areas where a baseline algorithm can learn to perform at least some meaningful action sequences already. Otherwise the Action Grammar may not extract meaningful subsequences. Has the algorithm been tried on sparse-reward games?


**Experience Assessment:**

I have read many papers in this area.

**Review Assessment: Checking Correctness Of Derivations And Theory:**

N/A

**Review Assessment: Checking Correctness Of Experiments:**

I assessed the sensibility of the experiments.

**Review Assessment: Thoroughness In Paper Reading:**

I read the paper at least twice and used my best judgement in assessing the paper.

---

> ### Author Response · Authors · 2019-11-13
> **Rebuttal with brief description of revised submission**
>
> Dear reviewer 3,
>
> We are very delighted and thankful for your assessment.
>
> We do agree that a detailed comparison with traditional HRL algorithms may be useful. During the development of this work we found it very challenging to do so under fair circumstances. Both Feudal Networks as well as h-DQNs require significant amounts of user-defined specifications/hyperparameters (such as sub-goals and hierarchy definition) and often may not be trained in a fully end-to-end fashion. Therefore, we decided to focus on an “ablation” comparison with DDQN and SAC with frame-skipping (i.e. the “naive” grammar of primitive actions that correspond to length 4 macro-actions).
>
> Regarding the use of macro-actions to improve sample efficiency: The baseline comparison as well as ablation studies try to address these issues and provide more insights. Could you be so kind as to clarify which aspects exactly remain unclear?
>
> Finally, yes, we do have preliminary results for a sparse rewards environment, namely for 5-disk Towers of Hanoi (see newly added appendix item F). The agent only receives a positive reward when achieving the final state. The results so far are only for the tabular case and without HAR or “Abandon Ship”. In our experience, the grammar macros not only propagate value information further back into the past, but also allow the agent to explore parts of the state space more efficiently.  We also believe that refining value estimates & efficient exploration are by no means orthogonal to each other. From the figure it also becomes apparent that the agent is able to amplify initial successful trajectories by encoding the action sequences in a grammar. Thereby, an action grammar provides an action representation & an effective form of memory.
>
> Best wishes & thank you for your time,
> The authors.

---

### Official Review · AnonReviewer2 · 2019-10-24
**Official Blind Review #2**

**Rating:** 1

**Review:**

This paper proposes the use of macro (i.e. aggregated) actions to address reinforcement learning tasks. The inspiration presented is from hierarchical grammar representations, and the method is tested on a subset of Atari games. The paper is overall well written, although many paragraph demonstrate a level of polish inadequate for a top level submission (repetitions, typos, etc.).
The main idea pursued in the work is extremely interesting and with likely important implications to recent DRL. The concept though is far from new: a quick search for "macro action reinforcement learning" points to a NIPS '99 paper from J. Randlov, though on top of my mind there should be even older work on the topic.
The perspective proposed of considering macro actions as atoms in a grammar is certainly intriguing, but the work proposed does not develop the concept. The macro actions are identified as patterns in action sequences, then built in straight hierarchies, without any distinction in type of atoms nor any rule to effectively make up a grammar.
The related work section is extremely lacking, with no work older than 2016. The introduction presents more background, marginally older than that (up to 2012), when grammars make for an entire field of study with decades of history.
The process is interesting and incorporates plenty of useful experience, which I would personally be glad to see published, although in the current context is insufficient as stand-alone contribution.
On a more personal note, I suggest the authors not to get discouraged, as I strongly believe such an avenue of research is worthy investigating. A few research questions which I think should be asked are:
- Are the agents actually learning to play the game? Just render the game with one of your best players. For example, achieving a score of 360 on Qbert barely takes constant down input, and the fact that comparable scores have been published before is of no support.
- Are long action sequences always useful? For Qbert for example an average move length of 8 learned from an initial, untrained policy, is sufficient to get off the screen consistently. While the Abandon Ship protocol can mitigate this, the RL exploration phase is done by random action selection (consider explicit exploration instead), and the action space grows fast from the small initial 6 actions with the addition of all the macro actions, possibly limiting the exploration capability and biasing towards the use of longer macro actions even when sub-optimal.
- Mitigate the claims. I would love to "eventually help make RL a universally practical and useful tool in modern society", but unfortunately I think no single contribution can today make such a claim.

**Experience Assessment:**

I have published in this field for several years.

**Review Assessment: Checking Correctness Of Derivations And Theory:**

I assessed the sensibility of the derivations and theory.

**Review Assessment: Checking Correctness Of Experiments:**

I assessed the sensibility of the experiments.

**Review Assessment: Thoroughness In Paper Reading:**

I read the paper thoroughly.

---

> ### Author Response · Authors · 2019-11-13
> **Rebuttal with brief description of revised submission**
>
> Dear reviewer 2,
>
> Thank you very much for your time, consideration and detailed review.
> We apologize for any writing errors and have corrected the mentioned mistakes (see updated submission document). We fully agree that the HRL sub-field of maco-actions dates back a lot longer than the literature cited in this submission. We have now revised the paper to address this; see section 2 with literature comparison. Here is a small excerpt from the new addition:
>
> “[...]Identification of suitable low level sub-policies poses a key challenge to HRL.
> Current approaches can be grouped into three main pillars:
> Graph theoretic (Hengst et al., 2002; Mannor et al., 2004; Simsek et al., 2004) and visitation-based (Stolle et al. 2002) approaches aim to identify bottlenecks within the state space. Bottlenecks are regions in the state space which characterize successful trajectories. This work, on the other hand, identifies patterns solely in the action space and does not rely on reward-less exploration of the state space. Furthermore, the proposed action grammar framework defines a set of macro-actions as opposed to full option-specific sub-policies. Thereby, it is less expressive but more sample-efficient to infer.
> Gradient-based approaches, on the other hand, discover parametrized temporally-extended actions by iteratively optimizing an objective function such as the estimated expected value of the log likelihood with respect to the latent variables in a probabilistic setting (Daniel et al., 2016) or the expected cumulative reward in a policy gradient context (Bacon et al., 2017; Smith et al., 2018). Grammar induction, on the other hand, infers patterns without supervision solely based on a compression objective. The resulting parse tree provides an interpretable structure for the distilled skill set.
> Furthermore, recent approaches (Vezhnevets et al., 2017; Florensa et al., 2017) attempt to split the goal declaration and goal achievement across different stages and layers of the learned architecture. Usually, the top level of the hierarchy specifies goals in the environment while the lower levels have to achieve such. Again, such architectures lack sample efficiency and easy interpretation. The context-free grammar-based approach, on the other hand, is a symbolic method that requires few rollout traces and generalizes to more difficult task-settings. . [...]”
>
> The reviewer brings up the concern that the inferred grammar is crudely flattened into a straight hierarchy. Thereby, the notion of production rules & sub-policies are lost. We have a different view on this: Firstly, all of the production rules may easily be recovered and identified during execution time. Thereby, the interpretation of a grammar-inferred rule of temporally-extended actions does not get lost. Furthermore, as reviewer 3 has highlighted, a deep hierarchy of policies is not required in order to obtain an effective action space of temporally-extended skills. We also want to highlight the additional novel introduction of “Hindsight Action Replay” which we believe to be of general interest to the HRL community of its own merit.
>
> All in all we hope to have addressed some of the productive comments and will attempt to address any further concerns and questions in future work. We thank the reviewer for all their input and advice, and hope that the body of follow-up work is going to come closer to our aspirations.
>
> Best wishes and again thank you for your time,
> The authors.

---

### Public Comment · ~Christopher_Leonard1 · 2019-10-23
**Concerns regarding the feasibility of the proposed method**

Dear Authors,


I have thoroughly read through the paper.  It is quite interesting.  I have a few questions regarding the feasibility of the proposed method.

First, according to the ablation study presented in Fig. 5, it seems that only HAR brings impact on the curves.  The other methods presented in Fig. 5, in contrast, do not seem to provide significant improvements (e.g., action balanced replay buffer, abandon ship, transfer learning, etc.).  This ablation study seems to reveal that the action balanced replay buffer, abandon ship, and transfer learning approaches discussed in the paper do not actually affect the performance.   I am wondering if the authors could provide stronger experimental results and more detailed explanation to justify the necessity of these approaches?

According to the paper, the proposed method only presents results for 0.3M.  For most contemporary DRL papers in the literature, the training procedure is typically performed for 10M or above, while 0.3M seems to be relatively short.  Please note that 0.3M time steps of training can not sufficiently represent the capability of a training method.  For many cases, learning curves rise after 1M or even 5M time steps (e.g., http://htmlpreview.github.io/?https://github.com/openai/baselines/blob/master/benchmarks_atari10M.htm).  For a fair comparison with the existing contemporary DRL approaches, I suggest the authors to extend the experimental results to 10M, which is more appropriate.

The third questions is regarding the action space.  Based on the statements presented in the paper, it seems that the action space of the agent grows with time (i.e., more and more macro actions are added to the action space of the agent.).  With a huge action space containing only  a constant number of primitive actions, it seems that the agent has a higher chance to select macro actions instead of its primitive actions.  I am wondering if the authors could provide the frequency of the macro actions used by the policy (after training)?  In addition, as the action space grows over time, the training difficulty also increases accordingly, indicating that the learning curves may become harder and harder to rise.   This is the other reason why I would like to request the authors to provide the training curves for up to 10M time steps to justify the effectiveness of the proposed methodology.  Moreover, it would be more appropriate to show the growing trend of the action space as well as the final size of it.

It would be nice if the authors could address my concerns regarding the proposed approaches and experimental results presented in this paper.

Thank you very much.


Best regards,
Christopher

---

### Decision · Program_Chairs · 2019-12-19

**Decision:**

Reject

**Comment:**

The topic of macro-actions/hierarchical RL is an important one and the perspective this paper takes on this topic by drawing parallels with action grammars is intriguing. However, some more work is needed to properly evaluate the significance. In particular, a better evaluation of the strengths and weaknesses of the method would improve this paper a lot.